# The Effect of Minimum and Maximum Air Temperatures in the Summer on Heat Stroke in Japan: A Time-Stratified Case-Crossover Study

**DOI:** 10.3390/ijerph18041632

**Published:** 2021-02-09

**Authors:** Shinji Otani, Satomi Funaki Ishizu, Toshio Masumoto, Hiroki Amano, Youichi Kurozawa

**Affiliations:** 1International Platform for Dryland Research and Education, Tottori University, Tottori 680-0001, Japan; 2Division of Health Administration and Promotion, Faculty of Medicine, Tottori University, Yonago 683-8503, Japan; chocolate.1423103@gmail.com (S.F.I.); tmasumoto@tottori-u.ac.jp (T.M.); h-amano@tottori-u.ac.jp (H.A.); kurozawa@tottori-u.ac.jp (Y.K.)

**Keywords:** heat stroke, minimum air temperature, ambulance transport, elderly

## Abstract

An increase in the global surface temperature and changes in urban morphologies are associated with increased heat stress especially in urban areas. This can be one of the contributing factors underlying an increase in heat strokes. We examined the impact of summer minimum air temperatures, which often represent nighttime temperatures, as well as a maximum temperature on a heat stroke. We collected data from the records of daily ambulance transports for heat strokes and meteorological data for July and August of 2017–2019 in the Tottori Prefecture, Japan. A time-stratified case-crossover design was used to determine the association of maximum/minimum air temperatures and the incidence of heat strokes. We used a logistic regression to identify factors associated with the severity of heat strokes. A total of 1108 cases were identified with 373 (33.7%) calls originating in the home (of these, 59.8% were the age of ≥ 75). A total of 65.8% of cases under the age of 18 were related to exercise. Days with a minimum temperature ≥ 25 °C had an odds ratio (95% confidence interval) of 3.77 (2.19, 6.51) for the incidence of an exercise-related heat stroke (reference: days with a minimum temperature < 23 °C). The odds ratio for a heat stroke occurring at home or for calls for an ambulance to the home was 6.75 (4.47, 10.20). The severity of the heat stroke was associated with older age but not with air temperature. Minimum and maximum air temperatures may be associated with the incidence of heat strokes and in particular the former with non-exertional heat strokes.

## 1. Introduction

Multiple factors contribute to the incidence of heat strokes; for example, environmental factors such as high temperatures and humidity, human factors such as dehydration, the presence of comorbidities, delayed heat acclimatization, aging and activity factors such as exercise and work in a hot environment [1]. In particular, climate change including global warming has contributed to the recent increase in the incidence of heat strokes. In 2018, the number of people transported to emergency rooms for a heat stroke in Japan was 95,137, the highest number on record. The number of deaths was 1581, the second highest on record [2,3]. Continued warming over time is expected [4] and up to 1.2 billion people worldwide will be at risk of a heat stroke each year by the year 2100 if global warming continues at this rate [5].

A number of studies have been conducted on the relationship between maximum temperatures and heat strokes, reporting a strong correlation [6,7,8,9,10,11]. We previously noted a strong correlation between maximum temperatures and the number of emergency transports for heat strokes and were able to quantify the risk by temperature [12]. However, to date, most studies have focused on the maximum temperature.

An urban heat island is a phenomenon wherein urban areas are warmer than the surrounding suburban and rural areas typically because of urban geometry and material properties, human activity and reduced natural vegetation. This phenomenon can intensify extreme climatic events [13]. Direct health effects involve elevated body temperatures particularly during heat waves [14,15]. Urbanization is associated with higher temperatures and the number of nights in Japan where the minimum temperature does not fall below 25 °C is increasing [16] not only in cities but also in rural areas [17].

Although large daily differences in temperatures exert negative effects on the human body [18], this may not apply in midsummer. During the summer months, the risk of a stroke and a transient ischemic stroke may increase if minimum temperatures remain high [19]. However, there are no reports on the association of minimum temperatures with the incidence of heat strokes. An association between persistent high temperature environments and the incidence of non-exertional heat strokes resulting from impaired physiological adaptation to heat stress, the so-called classical heat stroke, has been noted [1,20]. However, research on non-exertional heat strokes has received less attention than on exertional heat strokes [21,22]. Therefore, in this study, we used data on emergency transports to investigate the association between maximum and minimum temperatures and heat strokes. In addition, we examined the relationships of temperature for both non-exertional and exertional heat strokes.

## 2. Materials and Methods

### 2.1. Study Design and Sites

This study focused on the Tottori Prefecture, which is located on the Sea of Japan side of the Chugoku region in western Japan (Figure 1). The Tottori Prefecture is 3507 km^2^ in area and the population was 555,663 in October 2019. The percentage area and population of densely inhabited districts in the Tottori Prefecture (2005) are 1.35% (national average, 3.32%) and 34.1% (66.0%), respectively [23]. The Tottori Prefecture is sunny and warm in the summer and snowy in the winter because of the monsoon from the northwest. The penetration rate of air conditioners in the Tottori Prefecture (2014) was 87.1% (the national average, 86.4%) [24]. Out of 47 prefectures, the Tottori Prefecture had the highest number of heat stroke emergency transports per 100,000 population (May–September) in 2019 (76.69) and ranked eighth in 2018 (101.49) and 2017 (68.88) [25,26,27].

### 2.2. Ambulance Transport Data

We obtained data on daily ambulance transports for heat strokes from all 26 fire stations in the Tottori Prefecture for July and August of 2017, 2018 and 2019. These data were collected through the Department of Health and Welfare of the Tottori Prefecture, which identified a total of 1187 cases. We did not include the June and September data because the percentage of days with a minimum temperature of 25 °C or higher in these months was 2.8% and the number of heat stroke transports was 13.3% of the total. Each case record contained information on age and sex, the municipality to which the ambulance was called, the date and time of ambulance transport, the medical condition and the initial diagnosis. Furthermore, the record specified the situation surrounding the heat stroke; at home, at work (excluding farm work), exercise (both indoors and outdoors), outside excluding work, farm work, while watching an outside event, during tourist activities or other circumstances. The medical condition and initial diagnosis were determined by the emergency room physician based on the International Classification of Diseases 10th revision (T67: Effects of heat and light) [28] when the patient arrived at the hospital. The medical condition was classified as mild (cases not requiring hospitalization), moderate (non-mild or non-severe cases), severe (cases expected to be hospitalized for more than three weeks) or death. It should be noted that this severity classification is specific to emergency departments and differs from that of the Japanese Association for Acute Medicine Committee [29]. Cases included persons living outside the Tottori Prefecture and these were excluded from the study for reasons described below. The final count of cases included in the analysis was 1108.

### 2.3. Meteorological Data

Data on daily ambient air temperatures and the average relative humidity in the Tottori Prefecture were provided by the Japan Meteorological Agency. We used data from meteorological observation sites located in three areas; Tottori City as the eastern district (Iwami Town, Yazu Town, Wakasa Town and Chizu Town), Kurayoshi City as the central district (Hokuei Town, Misasa Town, Kotoura Town and Yurihama Town) and Yonago City as the western district (Sakaiminato City, Daisen Town, Hoki Town, Nanbu Town, Kofu Town, Nichinan Town, Hino Town and Hiezu Village) (Figure 1). We selected three of the nine regional weather stations in the Tottori Prefecture because the three cities where these stations are located account for 83.3% of the population and the location of heat stroke incidence is only known on a municipal scale. We used meteorological data from the observation point nearest the location to which the ambulance was called. As the Kurayoshi observation site did not provide relative humidity data, we used data from the Tottori observation site, which is closer in distance, as a substitute.

### 2.4. Statistical Analysis

We first performed descriptive analyses. Ambulance transport cases were grouped by sex and by age (children (<18 years), adults (18–64 years), semi-elderly (65–74 years) and elderly (≥75 years)). We also tallied cases by time and circumstances of ambulance transport.

We then analyzed the association of maximum air temperature on heat strokes using the time-stratified case-crossover approach. The case-crossover method is useful for assessing the association of transient exposures on the short-term risk of disease when only case data are available [30,31,32]. This design has also been used in recent years in studies such as those estimating the association between temperature factors and adverse health events [33,34]. In the case-crossover design, each case acts as its own control and exposure factors (in this study, the maximum and minimum ambient temperature of the day) are compared between cases and controls. The same days of all other weeks in the same month were selected as the control days (up to four control days per case were used) with the day of transport for the heat stroke as the event day. The reason for this was to eliminate as many confounding factors as possible by assuming that an individual’s life pattern was mostly determined by the day of the week [34,35,36,37]. Seventy-nine subjects living outside the Tottori Prefecture were excluded from the study because they were likely to be outside the Tottori Prefecture on the control days. We used a conditional logistic regression to estimate odds ratios (ORs) and 95% confidence intervals (CIs) for the association of maximum and minimum air temperatures on heat strokes for all cases and for each subgroup of age/gender. Furthermore, because heat strokes are broadly classified into exertional [38] and non-exertional (classical) [39] heat strokes, we classified cases occurring at home as a non-exertional heat stroke and cases associated with exercise as an exertional heat stroke.

Finally, we quantified the severity of mild, moderate, severe and death in order from 1 to 4 and conducted an ordered logistic regression analysis to identify factors associated with severity. *p* < 0.05 was considered significant. Analyses were conducted in SPSS version 24.0 (IBM-SPSS, Armonk, NY, USA).

### 2.5. Ethical Approval

The study was approved by the Tottori University Ethics Committee (No. 180626-069).

## 3. Results

### 3.1. Minimum, Average and Maximum Air Temperatures at the Study Sites

Table 1 summarizes the daily minimum, average and maximum air temperatures at observation sites located in the three cities of the Tottori Prefecture during the study period (three year average for 2017–2019). All values were higher than the cumulative annual average calculated for 1981–2010. Table 2 lists the total number of days with a minimum temperature of ≥25 °C, a maximum temperature of ≥30 °C and ≥35 °C at three observation sites (2017–2019).

### 3.2. Ambulance Transport Data

Table 3 lists the number of heat stroke cases by sex and age group. The mean age of the cases was 59.8 years and the median age was 69.0 years. The number of cases per 1000 residents was highest among elderly men (6.155), a 10-fold increase from the number among adult women (0.596).

An ambulance pickup for a heat stroke occurred mainly at home (33.7% of cases), work (20.9%), during outings (17.9%) and while exercising (15.6%). Ninety-seven percent of cases were mild or moderate with only four deaths (Table 4). A total of 59.8% of the heat stroke cases picked up at home were over 75 years of age and 70.4% of those were of moderate or greater severity.

Heat stroke cases were picked up the most in the hour from 11 am. Although not in large numbers, heat stroke cases were also transported at night and early in the morning. 64% of all heat stroke cases in the 12 h after 8 pm were picked up at home (Figure 2).

### 3.3. Case-Crossover Analysis

#### 3.3.1. Maximum Air Temperature

We investigated the risk of a heat stroke on days when the maximum air temperature exceeded 30 °C. The risk of a heat stroke increased when the maximum air temperature exceeded the reference temperature (<30 °C; Figure 3). The overall OR (95% CI) was 6.26 (4.67, 8.38) for 30–34.9 °C, 12.24 (8.85, 16.93) for 35–36.9 °C and 17.55 (12.16, 25.32) for ≥ 37 °C. The risk of a heat stroke was higher in individuals aged <18 years than in other age groups and in women. Engaging in exercise was more likely to result in a heat stroke than other activities.

#### 3.3.2. Minimum Air Temperature

We investigated the risk of a heat stroke on days when the minimum air temperature exceeded 23 °C. The risk of a heat stroke increased when minimum air temperatures exceeded the reference temperature (<23 °C; Figure 4). The overall OR (95% CI) was 3.22 (2.58, 4.01) for 23–24.9 °C and 5.36 (4.30, 6.69) for ≥25 °C. When air temperatures were 23–25 °C, the risk of a heat stroke was higher in individuals aged 18–64 years and in women. The risk was higher in individuals who were at home at the time that the ambulance was called.

### 3.4. Relationship between Air Temperature and Heat Stroke Severity

We investigated the risk analyzed heat stroke severity using two models (Table 5). In the first model, we used age, sex, maximum air temperature, relative humidity, a heat stroke occurring at home and a heat stroke associated with exercise as independent variables. In the second model, we replaced the maximum air temperature with a minimum air temperature. Only age was associated with the severity in either model (*p* < 0.001). Being at home when calling for medical transport showed a tendency toward association but did not reach statistical significance in either model (*p* = 0.070 and 0.098). Likewise, minimum temperatures showed a tendency toward association but did not reach statistical significance (*p* = 0.086).

## 4. Discussion

In the present study, we found that the daily minimum air temperature during summer may influence the incidence of a heat stroke. This effect does not exceed that of maximum temperatures but is probably not negligible. Typically, the lowest temperature of the day is often recorded around dawn and therefore high minimum temperatures are indicative of a hot night [40]. Higher nighttime temperatures have been reported to lead to sleep deprivation with the greatest impact seen in the summer months among low-income and elderly individuals [41]. Sleep deprivation has been cited as a risk factor for a heat stroke and research on the health effects of nighttime temperatures associated with global warming and the heat island phenomenon will become increasingly important [38].

We found that the risk of a non-exertional heat stroke when the minimum temperature was 25 °C or higher was approximately twice the risk associated with exercise. In general, a large temperature gradient during the day can exert adverse health effects such as mortality from cardiovascular disease, respiratory disease and stroke [18,42]. In contrast, cerebrovascular events in the summer have been reported to increase on days with smaller differences in air temperature or higher minimum air temperatures [19]. In the present study, we did not find a clear relationship between the temperature and the severity of the heat stroke, perhaps because the initial diagnosis could change and we had no information on the medical history and pre-existing conditions of cases. Although the number of cases in this study was small, the findings suggest that hot nights may be associated with the severity of heat strokes especially non-exertional heat strokes.

Japan has one of the highest percentages of older individuals in the world [43] and just over half (58%) of cases were aged 65 years and above in the present study. Therefore, we stratified those aged 65 years and over into two groups. The elderly are at a higher risk for heat strokes because of declining physiological and cognitive functions [43,44,45,46,47] and also in this study, the severity of the heat stroke was associated with older age. Furthermore, a non-exertional heat stroke is common in the elderly [48] who may not use air conditioning even if it is available [45,46,47]. The risk of a heat stroke associated with increasing temperatures was not as high as in the younger group presumably because the elderly are more vulnerable to environmental exposures even at temperatures below 30 °C.

We found that women were more susceptible to heat strokes than men especially when the maximum temperature was 37 °C or higher. This finding is supported by the literature [49]. A high body mass index is a risk factor for heat stroke [50] and women tend to have a higher body mass index than men [50,51]. However, there are no physical data on the subjects’ height or weight in our study and further research is needed to determine the relationship between heat strokes and body composition.

We only used the air temperature as a meteorological indicator in the case-crossover analysis and did not include relative humidity. Our previous study did not find an association with heat stroke transport [12] and we also found no association with heat stroke severity in the present study. One of the reasons is that the relative humidity in the target area remained extremely high in July and August. In fact, according to official data from the Japan Meteorological Agency, the average relative humidity ± standard deviation for the study period was 75.2 ± 8.6% at the Tottori observation site and 75.0 ± 9.2% at the Yonago observation site. Therefore, the effect of humidity may not have diminished. Although wet-bulb globe temperature (WBGT) [52] and apparent temperature (AT) [9,53] are standardized general indicators in the studies of heat strokes we did not use both because these calculations require data on relative humidity. In addition, radiant heat values are necessary for WBGT calculations [51] and wind-speed data for AT [54] but these values are very localized [55]. 

Several limitations should be noted. First, we have data on the location and time of the emergency transport; however, we do not know the exact location and time that the heat stroke developed. The heat stroke could have developed hours earlier and elsewhere in a few cases. We also did not use time series data of air temperatures but simply the daily minimum and maximum temperature data for analysis. It is likely that some of the highest temperatures may have been recorded after a heat stroke event. Moreover, the meteorological data were collected at representative locations, perhaps even 30 km from the location of the case, and therefore may not reflect the actual temperatures in each individual case. Second, ambulance transport data do not include heat stroke cases that arrived at emergency departments on their own. Third, the diagnosis of a heat stroke was made by a clinician at the time of the initial examination and may have ultimately been changed. Fourth, we had no access to the medical histories of cases, precluding the analysis of the influence of pre-existing conditions. As older individuals have more comorbidities, the risk in the elderly may be overestimated. However, ambulance transport data are frequently used because they are available and may be suitable for establishing a real-time surveillance system [56]. Finally, the use of a case-crossover design for statistical analysis may also be problematic [57,58]. It has been pointed out that this design has less power when compared with a Poisson regression analysis [57]. However, because the case-crossover design was thought to clear up the results of the subgroup analysis, we used this method in current study [57]. For a more detailed and universal assessment of environmental influences on the incidence of heat strokes, the effect of a lag in the day or two days before calling an ambulance, actual morbidity and post-hospitalization outcomes as well as the socioeconomic background of patients should be included in analyses.

## 5. Conclusions

Higher minimum and maximum daily temperatures are associated with the incidence of heat strokes in the summer, especially non-exertional heat strokes.

## Figures and Tables

**Figure 1 ijerph-18-01632-f001:**
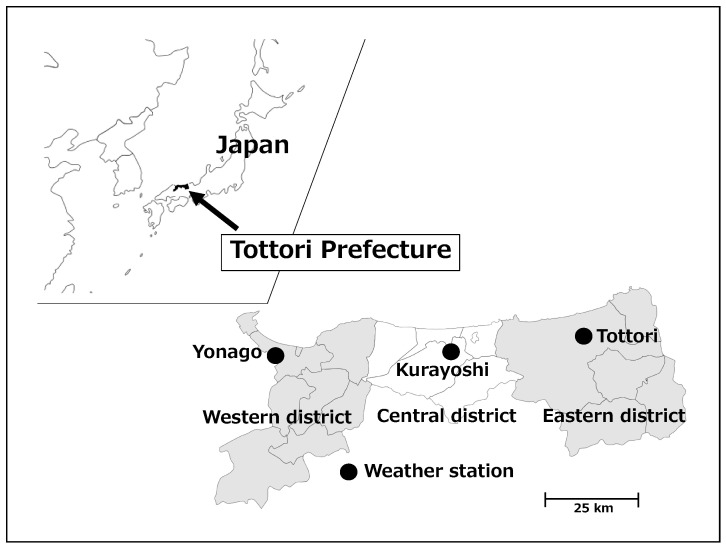
Location of the Tottori Prefecture and meteorological observation sites. The eastern district (right dark area) includes five, the central district (light area) includes five and the western district (left dark area) includes nine municipalities.

**Figure 2 ijerph-18-01632-f002:**
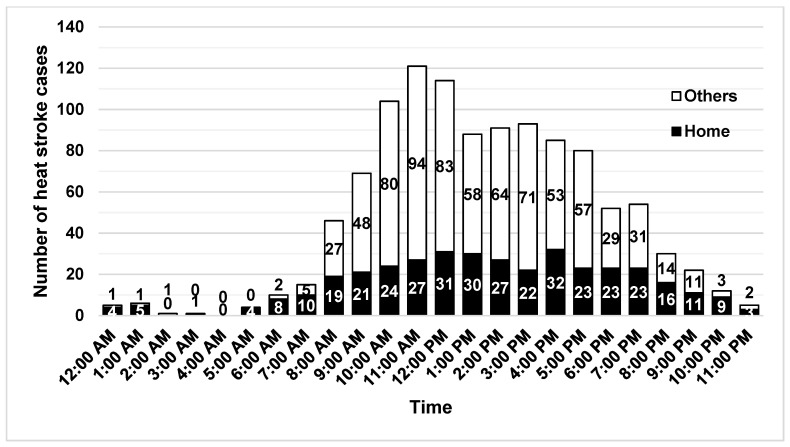
Pickup time for heat stroke cases. Black bars indicate the number of pickups at home and white bars indicate the others.

**Figure 3 ijerph-18-01632-f003:**
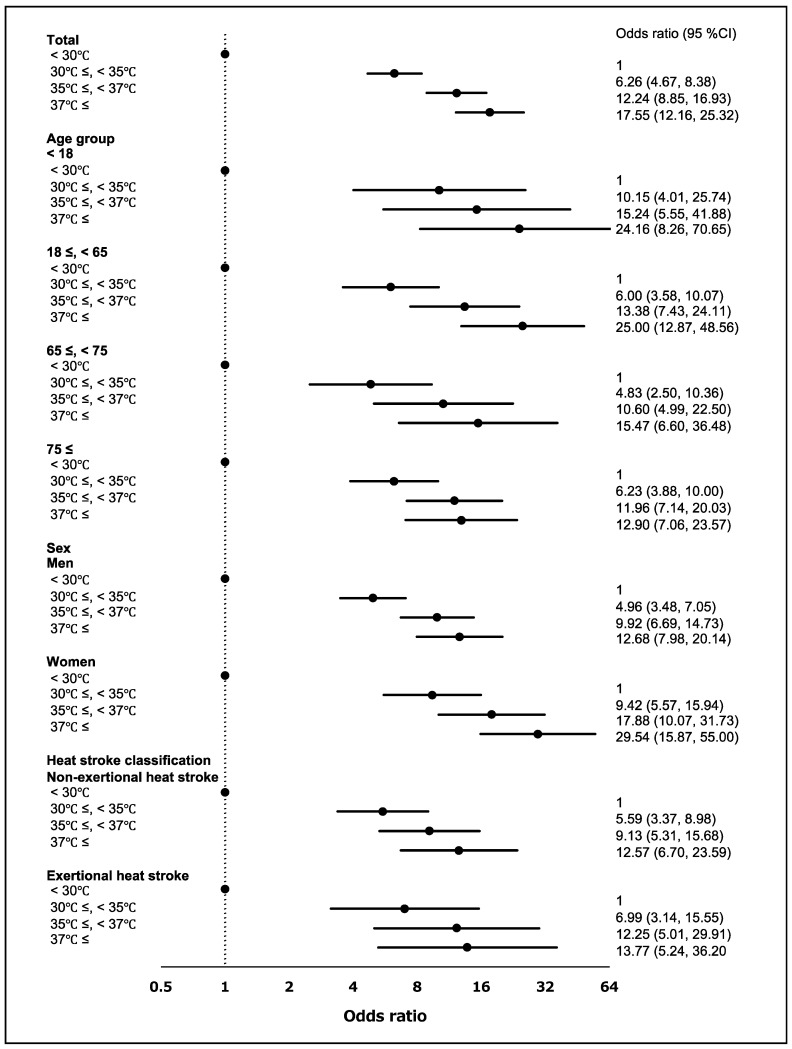
Risk of developing a heat stroke by maximum air temperature and stratified by age, sex and heat stroke classification. The black dots mean the odds ratio. Numbers in parentheses are the 95% confidence intervals (95% CI).

**Figure 4 ijerph-18-01632-f004:**
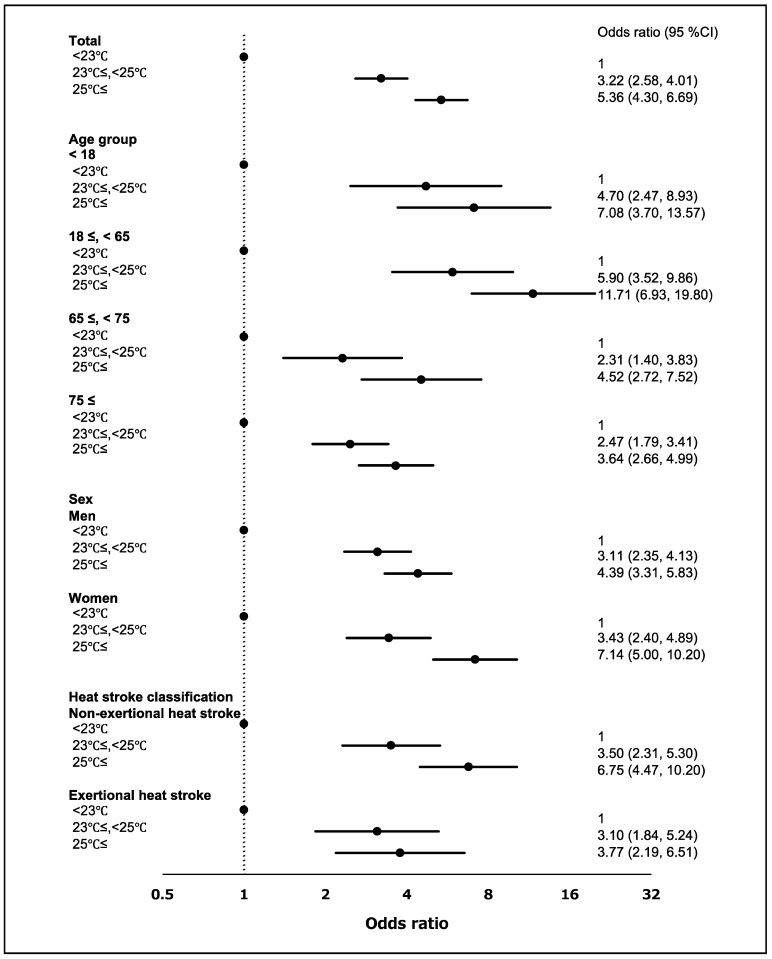
Risk of developing a heat stroke by minimum air temperature and stratified by age, sex and heat stroke classification. The black dots mean the odds ratio. Numbers in parentheses are the 95% confidence intervals (95% CI).

**Table 1 ijerph-18-01632-t001:** Daily minimum, average and maximum air temperatures (°C) in three districts of the Tottori Prefecture during the study period (three year average for 2017–2019).

	July			August		
	Minimum Value	Average Value	Maximum Value	Minimum Value	Average Value	Maximum Value
Tottori	23.6 (22.0)	27.3 (25.7)	32.1 (30.4)	23.9 (22.9)	28.0 (27.0)	33.5 (32.2)
Kurayoshi	22.8 (21.7)	26.0 (24.9)	29.8 (28.8)	22.9 (22.4)	26.6 (26.0)	30.8 (30.2)
Yonago	23.9 (22.3)	27.3 (25.6)	31.5 (29.6)	24.3 (23.3)	28.1 (26.9)	32.8 (31.2)

Figures in parentheses indicate the cumulative annual average calculated for 1981–2010.

**Table 2 ijerph-18-01632-t002:** Days with a minimum temperature of ≥25 °C, a maximum temperature of ≥ 0 °C and ≥35 °C (2017–2019).

	July			August		
	T_min_25 °C	T_max_30 °C	T_max_35 °C	T_min_25 °C	T_max_30 °C	T_max_35 °C
Tottori	19	70	36	31	81	36
Kurayoshi	11	49	2	11	57	3
Yonago	32	65	19	41	77	25

T_min_25 °C: a day with a minimum temperature of ≥25 °C, T_max_30 °C: a day with a maximum temperature of ≥30 °C, T_max_35 °C: a day with a maximum temperature of ≥35 °C.

**Table 3 ijerph-18-01632-t003:** Number of heat stroke cases (2017–2019).

Age Group	Men	Women	Total
<18	83 (1.901)	85 (2.043)	168 (1.970)
18≤, <65	216 (1.488)	86 (0.596)	302 (1.043)
65≤, <75	129 (3.27)	58 (1.343)	187 (2.242)
75≤	210 (6.155)	241 (4.086)	451 (4.845)
Total	638 (2.401)	470 (1.769)	1108 (1.994)

Numbers in parentheses are cases per 1000 population as of October 2019.

**Table 4 ijerph-18-01632-t004:** Location or activity reported by heat stroke cases and stratified by age and severity.

	Home	Work	Exercise	Outing	Farm Work	Event	Tourism	Others	Total
Severity									
Mild	139	115	111	104	35	10	8	1	523
Moderate	219	113	62	90	55	6	7	1	553
Severe	14	3	0	4	7	0	0	0	28
Death	1	1	0	0	2	0	0	0	4
Total	373	232	173	198	99	16	15	2	1108
Age group									
<18	7	11	110	26	0	11	3	0	168
	(42.9%)	(36.4%)	(35.5%)	(34.6%)	(–)	(36.4%)	(33.3%)	(–)	(35.7%)
18≤, <65	60	138	34	49	9	4	8	0	302
	(46.7%)	(48.6%)	(35.3%)	(38.8%)	(55.6%)	(25.0%)	(50%)	(–)	(45.0%)
65≤, <75	83	34	7	36	22	1	3	1	187
	(55.4%)	(38.2%)	(14.3%)	(61.1%)	(54.5%)	(100%)	(66.7%)	(100%)	(52.4%)
75≤	223	49	22	87	68	0	1	1	451
	(70.4%)	(67.3%)	(45.5%)	(50.6%)	(69.1%)	(–)	(0%)	(0%)	(64.5%)
Total	373	232	173	198	99	16	15	2	1108
	(62.7%)	(50.4%)	(35.8%)	(47.5%)	(64.6%)	(37.5%)	(46.7%)	(50%)	(52%)

The situation surrounding picking up heat stroke cases: at home (Home), at work excluding farm work (Work), exercise both indoors and outdoors (Exercise), outside excluding work (Outing), farm work (Farm Work), while watching an outside event (Event), during tourist activities (Tourism) or other circumstances (Others). Figures in parentheses indicate the percentage of mild, moderate, severe or death cases.

**Table 5 ijerph-18-01632-t005:** Ordered logistic regression factors affecting the severity of a heat stroke.

**Model 1**					
**Variable**	**Coefficient**	**Standard error**	***p* Value**	**95% Confidence Interval**	
				**Lower**	**Upper**
Age	0.014	0.003	<0.001	0.009	0.019
Sex	−0.027	0.124	0.829	−0.270	0.217
Maximum air temperature	0.041	0.034	0.232	−0.026	0.108
Relative humidity	0.007	0.012	0.561	−0.016	0.030
Home	0.254	0.140	0.070	−0.021	0.529
Exercise	−0.261	0.198	0.187	−0.649	0.127
**Model 2**					
**Variable**	**Coefficient**	**Standard error**	***p* Value**	**95% Confidence Interval**	
				**Lower**	**Upper**
Age	0.015	0.003	<0.001	0.009	0.020
Sex	−0.018	0.124	0.886	−0.262	0.226
Minimum air temperature	0.068	0.040	0.086	−0.010	0.146
Relative humidity	0.005	0.010	0.630	−0.014	0.023
Home	0.233	0.141	0.098	−0.043	0.509
Exercise	−0.237	0.199	0.234	−0.626	0.153

## Data Availability

The data presented in this study are available on request from authors.

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
