# Peer review of "The Effect of Minimum and Maximum Air Temperatures in the Summer on Heat Stroke in Japan: A Time-Stratified Case-Crossover Study"

_ijerph, 2021, doi:10.3390/ijerph18041632_

Round 1
Reviewer 1 Report
This study aims to investigate the impact of air temperature on heat-stroke in Tottori Prefecture, Japan. By using meteorological data and incidences of heat-stroke (ambulance transport) from 2017 to 2019, this case-crossover study applied logistic regression analysis to examine the interconnection between air temperature and heat-stroke. After checking the manuscript, we appreciate the authors’ efforts to provide some improvements in this current version. However, because I haven't found a point-by-point response for each of the comments, to follow up on the previous review, the minor comments below may need a response.
- A previous study confirmed the effect of obesity/overweight on heat disorders (Chung and Pin. 1996. Obesity and the occurrence of heat-disorders. Mil Med. 1,161 (12): 739-42. PMID: 8990832). Connecting to this study, what are the authors' opinions?
If obesity is one of the influencing factors, it would be better if this can be briefly and clearly addressed as a study limitation and also a suggestion for future studies.
Designating to Figure 3, just an open question. I am a bit confused about how this study analyzes the risk of developing heat-stroke at temperatures ≥35°C, considering the results of the statistical analysis in Table 1 show that the highest temperature of the three studied regions is around 34°C (to be precise 33.5 C for the maximum temperature in Tottori).
Author Response
Dear Reviewer 1
Thank you very much for reviewing our paper and giving us some important suggestions. We would like to explain some of your points again.
To the first point:
This point was pointed out by the reviewer at the first time and we have added to our previous revise version. Since this study does not include physical information of each case, we cannot actually assess the risk by obesity or BMI. We have included this point as a future issue in lines 229-231, and we hope you will check it.
To the second point:
The reviewer pointed this out to us at the time of the first submission, so we thought we would improve it by creating a new Table 2 in our revise version. Table 1 shows the average minimum temperature, average daily temperature, and average maximum temperature for the observation period at each site. Table 2 shows the number of days with minimum temperatures ≥25°C, and maximum temperatures ≥30°C and ≥35°C by month and region. Table 2 shows, for example, that there are 36 days in July when the temperature is ≥35℃ in Tottori City. There is a total of 121 days with a maximum temperature of ≥35°C, so it is possible to calculate the risk when the temperature is ≥35°C or higher. We hope this answers your question.
Again, thank you for giving us the opportunity to strengthen our manuscript with your valuable comments and queries. We have worked hard to incorporate your feedback and hope that these revisions persuade you to accept our submission.
Sincerely yours,
Shinji Otani

Reviewer 2 Report
Dear authors,
thank you for your efforts to strengthen your manuscript.
I learned from your revisions that you did not have the temperature time series for 24hours/7days since you could not relate exact temperature to alert time.
Please indicate in the manuscript that you only had the min/max and calculated average values for every day.
Please correct typos: line 141+142 mUximum
Please check in your diagramm if it is 12:00 pm or 12:00 am
Author Response
Dear Reviewer 2
Thank you for your careful review of our paper. We have included the points you raised as part of the limitation (Lines 245-247), and corrected the spelling mistakes.
Again, thank you for giving us the opportunity to strengthen our manuscript with your valuable comments and queries. We have worked hard to incorporate your feedback and hope that these revisions persuade you to accept our submission.
Sincerely yours,
Shinji Otani
Reviewer 3 Report
The paper, after this further round of revision, has been improved and now it is ready to be published.
Author Response
Dear Reviewer 3
Thank you very much for your careful review of our paper. We have taken into account the points you have raised and have made improvements. We hope that these corrections will be satisfactory to you.
Sincerely yours,
Shinji Otani

This manuscript is a resubmission of an earlier submission. The following is a list of the peer review reports and author responses from that submission.
Round 1
Reviewer 1 Report
The paper analyses the correlation of heatstroke and temperature by using data collected over three years in a prefecture of Japan. As such the paper's topic falls within the ones suitable for this journal. The paper is generally well written and includes detailed information on methodology in order to replicate the study. The paper needs some improvements before it will be ready to be published. The main issues are the following:
- Title: The title needs to be revised. From the title, it seems that the paper aims at assessing the relationship between minimum air temperature and heat stroke. In the paper, on the contrary, is assessed not only the relationship between heat stroke and minimum air temperature, but also the one between maximum air temperature and heat stroke.
- Introduction: The authors might give a more detailed overview of similar studies carried out recently (there are several published in the last year) and highlight what is the original contribution of this paper in comparison with previous researches.
- Methodology: The authors should clarify why only air temperature was considered in this study, and not - for instance - apparent temperature. In several papers it has been highlighted how there is a strong relationship between apparent temperature and heat-related mortality and morbidity, as also relative humidity has an effect on reaction mechanisms of human body to high temperature levels. The authors should discuss this and demonstrate why air temperature has been considered in their study, instead.
Author Response
Dear Reviewer,
Thank you very much for providing important insights. In the following sections, you will find our responses to each of your points and suggestions. We are grateful for the time and energy you expended on our behalf.
--
1. Title: The title needs to be revised. From the title, it seems that the paper aims at assessing the relationship between minimum air temperature and heat stroke. In the paper, on the contrary, is assessed not only the relationship between heat stroke and minimum air temperature, but also the one between maximum air temperature and heat stroke.
-Thank you for your point about the title. Initially, only the minimum air temperature was assessed, and as a result, the title was left unchanged. The title was changed to "Effect of minimum and maximum air temperatures in summer on heat stroke in Japan: a time-stratified case-crossover study".
2. Introduction: The authors might give a more detailed overview of similar studies carried out recently (there are several published in the last year) and highlight what is the original contribution of this paper in comparison with previous researches.
Thank you for pointing that out. This study focuses on minimum temperatures and also mentions non-exertional heat stroke. We have added references to highlight these points and have also restructured and revised the text (Introduction).
3. Methodology: The authors should clarify why only air temperature was considered in this study, and not - for instance - apparent temperature. In several papers it has been highlighted how there is a strong relationship between apparent temperature and heat-related mortality and morbidity, as also relative humidity has an effect on reaction mechanisms of human body to high temperature levels. The authors should discuss this and demonstrate why air temperature has been considered in their study, instead.
Initially, we considered assessing the effects of relative humidity and WBGT and/or apparent temperature, but in the end we only assessed the temperature. The main reason is that of the three meteorological stations, the Kurayoshi observation site is small in size and does not record humidity data. As a result, we were unable to calculate the index including humidity. We have made additions and corrections to include other points (lines 229-235).
--
Again, thank you for giving us the opportunity to strengthen our manuscript with your valuable comments and queries. We have worked hard to incorporate your feedback and hope that these revisions persuade you to accept our submission.
Sincerely yours,
Shinji Otani
Reviewer 2 Report
Dear authors,
I`d like to applaude for your efforts in conducting this study on the effects of ambient temperatures on the incidence of heatstroke. The presented topic is of global interest. However, I would like to recommend some improvements and beg you for some clarifications. The writing style is of high quality but the use of sentence connectors might help ease the reading flow.
Title:
Please add "maximum" temperature or do not mention "maximum" or "minimum" at all. You analysed both temperatures, so it is misleading to only pinpoint "minimum".
Abstract:
-Same problem as in the title. "We examined the impact of summer minimum air temperatures" is not sufficient.
-Since the incidence for heatstroke at home or while exercising is influenced by age (and the respective size of the population), a global odds ratio does not present useful information. I would recommend describing the distinctive patterns for each group. E.g. "While younger patients developed more likely exertional heat stroke, older patients were more prone for heatstroke at home..."
Methods:
-Please indicate the definition of heatstroke and add the severity scale as an addendum
- Setting: What is the level of urbanisation in Tottori Prefecture? Are houses regularly equipped with air conditioning?
-Ambulance Transport Data: Why were 79 cases excluded?
-Meteorological data: Please indicate that ONLY three observation sites where used. Please describe the level of urbanization around the sites.
-Statistical Analysis: Please explain in plain language how a case-crossover analysis is performed. How can each case serve as its own control?
-What kind of time trends do you mean?
-"To avoid bias due to time trends, we selected control periods from the same day of the week in the same month"--> Do you mean: the same weekday of another week in the same month? Please rewrite.
Results:
-Table 1: Please indicate if you constantly used the average values for min/average/max of the years 2017-2019. Or if you used the absolute values of any year 2017-2019 for min/max and the average 2017-2019 for the average temperature.
-Table 3: Please add the relative percentages within each age group. Please consider adding an extended table 3 as an addendum in which you combine the age group and the severity level. It seems to be important to analyse what the severity levels are e.g. for young or old people depending on the location of ambulance pick-up.
-Time of ambulance alert/pick-up time is missing. Please include an analysis of the pick-up time in relation to min/max-temperatures. The minimum requirement is to analyse pickup-time between close to max-temperature vs. close to min-temperature.
Discussion:
Overall the discussion is sound. Most important points are discussed. I would recommend highlighting the different incidence pattern within each age group (cf. comments on abstract).
Summed up, the manuscript would profit from extended analysis of the data and the consideration of the above mentioned recommendations.
I would like to encourage the authors to consider resubmitting an improved version of the manuscript.
Author Response
Dear Reviewer,
Thank you very much for providing important insights. In the following sections, you will find our responses to each of your points and suggestions. We are grateful for the time and energy you expended on our behalf.
--
Title:
Please add "maximum" temperature or do not mention "maximum" or "minimum" at all. You analysed both temperatures, so it is misleading to only pinpoint "minimum".
-Thank you for your point about the title. Initially, only the minimum air temperature was assessed, and as a result, the title was left unchanged. The title was changed to "Effect of minimum and maximum air temperatures in summer on heat stroke in Japan: a time-stratified case-crossover study".
Abstract:
1. Same problem as in the title. "We examined the impact of summer minimum air temperatures" is not sufficient.
-We added the words "as well as maximum temperature" to the original to match the title and text.
2. Since the incidence for heatstroke at home or while exercising is influenced by age (and the respective size of the population), a global odds ratio does not present useful information. I would recommend describing the distinctive patterns for each group. E.g. "While younger patients developed more likely exertional heat stroke, older patients were more prone for heatstroke at home..."
-We had originally described the point you made, but we had to delete it due to word count limitations. Your suggestion is valid, and we added it again (line 22).
Methods:
1. Please indicate the definition of heatstroke and add the severity scale as an addendum
-The definition of heat stroke is based on ICD-10, and the severity of heat stroke is based on Tottori Prefecture's emergency record procedures. We added a note on these (Lines 81-85).
2. Setting: What is the level of urbanisation in Tottori Prefecture? Are houses regularly equipped with air conditioning?
-We added these information in “2.1. Study Design and Sites” (Lines 63-67).
3. Ambulance Transport Data: Why were 79 cases excluded?
-The reason for this is related to the case-crossover design and we have provided an explanation within the section "2.4 Statistical Analysis" (Lines 110-111).
4. Meteorological data: Please indicate that ONLY three observation sites where used. Please describe the level of urbanization around the sites.
-Three of the nine regional weather stations in Tottori Prefecture were selected because these three stations cover 83.3% of the population and the location of heat stroke is only known on a municipal scale. We added this point (lines 95-97). We also corrected “the address to which the ambulance was called” in 2.2 Ambulance Transport Data to "the municipalities…"(Line 76).
5. Statistical Analysis: Please explain in plain language how a case-crossover analysis is performed. How can each case serve as its own control?
6. What kind of time trends do you mean?
7. "To avoid bias due to time trends, we selected control periods from the same day of the week in the same month"--> Do you mean: the same weekday of another week in the same month? Please rewrite.
-Thank you for your suggestion. We summarize these (5-7). In the case-crossover design, each case acts as its own control, and exposure factors (in this study, the maximum and minimum ambient temperature of the day) are compared between cases and controls. The same days of other weeks in the same month were selected as the control days, with the day of transport for heat stroke as the event day. The reason for this is to eliminate as many confounding factors as possible by assuming that an individual's life pattern is mostly determined by the day of the week. 79 out-of-prefecture residents were excluded from the study because they were likely to be outside of Tottori Prefecture on the control days. We revised the description of the case-crossover design in "2.4 Statistical Analysis" to make the point clearer to readers (lines 104-111).
Results:
1. Table 1: Please indicate if you constantly used the average values for min/average/max of the years 2017-2019. Or if you used the absolute values of any year 2017-2019 for min/max and the average 2017-2019 for the average temperature.
-Thank you for pointing that out. These are averages for three years, and we added them to the table explanation.
2. Table 3: Please add the relative percentages within each age group. Please consider adding an extended table 3 as an addendum in which you combine the age group and the severity level. It seems to be important to analyse what the severity levels are e.g. for young or old people depending on the location of ambulance pick-up.
-We redesigned the table. Since the number of severe cases and deaths are small, we included the total percentage of moderate, severe and death cases (Table 4).
3. Time of ambulance alert/pick-up time is missing. Please include an analysis of the pick-up time in relation to min/max-temperatures. The minimum requirement is to analyse pickup-time between close to max-temperature vs. close to min-temperature.
-As you pointed out, we thought the distribution of pickup times was necessary, so we created a new diagram. However, we did not have data on the times when the maximum and minimum temperatures were recorded, so we could not analyze the detailed association with heat stroke pickup times.
Discussion:
Overall the discussion is sound. Most important points are discussed. I would recommend highlighting the different incidence pattern within each age group (cf. comments on abstract).
-As mentioned above, we added a sentence in the Abstract (line 22). We rewrote some of the "Discussion" sections to take this into account (lines 215-222).
--
Again, thank you for giving us the opportunity to strengthen our manuscript with your valuable comments and queries. We have worked hard to incorporate your feedback and hope that these revisions persuade you to accept our submission.
Sincerely yours,
Shinji Otani
Reviewer 3 Report
This study aims to investigate the impact of air temperature on heat-stroke in Tottori Prefecture, Japan. By using meteorological data and incidences of heat-stroke (ambulance transport) from 2017 to 2019 (i.e. July and August), this case-crossover study applied logistic regression analysis to examine the interconnection between air temperature and heat-stroke. The first finding showed that heat-stroke was associated with age (elderly) but not significant with air temperature. Furthermore, air temperature (both summer minimum and maximum) maybe have an association with heat-stroke incidence and in particular non-exertional heat-stroke. To support the publication of this study, some comments below need to be considered.
- In general, please pay attention to typos or technical writing – proofread might helpful.
- Referring to the topics and study objective described in the abstract, this study aims to investigate the impact of minimum air temperature on heat-stroke (Lines 15-16). However, after checked the manuscript, explanation in the introduction (Lines 50-51), materials and methods - statistical approach (Lines 87-88), and also the results section confirmed that this study not only examined the impact of minimum temperature but also maximum temperature. Please clarify and make all explanations more consistent.
- In the introduction part, a brief and clear explanation related to the background of the heat-stroke condition in the study area (Tottori Prefecture) could be considered to reinforce the urgency of this study.
- Why does this study only use data for July and August? Is it related to the summer minimum and maximum? It will be obvious if this point could describe in detail in the materials and methods section.
- I suggest that the authors could add information about how many stations were used to observe meteorological data (in sub-chapter 2.3. Meteorological Data). Is it only 3 stations as shown in Figure 1? If so, is it not too few to estimate temperature conditions in the study area?
- A previous study confirmed the effect of obesity/overweight on heat disorders (Chung and Pin. 1996. Obesity and the occurrence of heat-disorders. Mil Med. 1,161 (12): 739-42. PMID: 8990832). Connecting to this study, what are the authors' opinions?
- (Figure 1) What differentiates between dark areas (Yonago - Tottori) and light area (Kurayoshi) shown on the map?
- Designating to Figure 2, it would be clear if the authors could describe the stratified analysis related to temperature classifications (<30°C; 30-34°C; 35-36°C; 37+°C).
Maybe this is an open question. I am a bit confused about how this study analyzes the risk of developing heat-stroke at temperatures ≥35°C, considering the results of the statistical analysis in Table 1 show that the highest temperature of the three studied regions is around 34°C (to be precise 33.5 C for the maximum temperature in Tottori).
- (Lines 158-160) “Higher nighttime temperatures have been reported to lead to sleep deprivation, with the greatest impact seen in the summer months among low-income and elderly individuals”. Regarding this statement, is it possible to consider socioeconomic conditions (e.g. income, education, etc.) as the potential covariates for model adjustment related to the impact of air temperature on heat-stroke? Perhaps this can also be noted as a limitation of this study.
- In strengthening the results of this study, the wet-bulb globe temperature (WBGT) index as a measure of heat stress can also be considered.
Author Response
Dear Reviewer,
Thank you very much for providing important insights. In the following sections, you will find our responses to each of your points and suggestions. We are grateful for the time and energy you expended on our behalf.
--
1. In general, please pay attention to typos or technical writing – proofread might helpful.
-We made several corrections to make the flow of the text easier to understand.
2. Referring to the topics and study objective described in the abstract, this study aims to investigate the impact of minimum air temperature on heat-stroke (Lines 15-16). However, after checked the manuscript, explanation in the introduction (Lines 50-51), materials and methods - statistical approach (Lines 87-88), and also the results section confirmed that this study not only examined the impact of minimum temperature but also maximum temperature. Please clarify and make all explanations more consistent.
-Thank you for your suggestion. We changed the title to "Effect of minimum and maximum air temperatures in summer on heat stroke in Japan: a time-stratified case-crossover study".
3. In the introduction part, a brief and clear explanation related to the background of the heat-stroke condition in the study area (Tottori Prefecture) could be considered to reinforce the urgency of this study.
-Thank you for your valuable feedback. Tottori Prefecture has the highest number of emergency heat stroke cases per population among the 47 prefectures in Japan. We added this point at the beginning of "2.1. Study Design and Sites " (Line 67-69).
4. Why does this study only use data for July and August? Is it related to the summer minimum and maximum? It will be obvious if this point could describe in detail in the materials and methods section.
-This study focuses on minimum temperatures in summer. In fact, during the observation period of this study, there were only 2 days/180 days in June (the total number of days in the three observation sites over the three years) when the minimum temperature was 25°C or higher, and 8 days in September. Since the number of heat-stroke transported in June and September was small (13% of the total number of heat stroke), we only analyzed the data for July and August. We added this to "2.2 Ambulance Transport Data" (Lines 73-76).
5. I suggest that the authors could add information about how many stations were used to observe meteorological data (in sub-chapter 2.3. Meteorological Data). Is it only 3 stations as shown in Figure 1? If so, is it not too few to estimate temperature conditions in the study area?
- Three of the nine regional weather stations in Tottori Prefecture were selected because these three stations cover 83.3% of the population and the location of heat stroke is only known on a municipal scale. We added this point (lines 95-97). We also corrected “the address to which the ambulance was called” in 2.2 Ambulance Transport Data to "the municipalities…"(line 76).
6. A previous study confirmed the effect of obesity/overweight on heat disorders (Chung and Pin. 1996. Obesity and the occurrence of heat-disorders. Mil Med. 1,161 (12): 739-42. PMID: 8990832). Connecting to this study, what are the authors' opinions?
-Our paper mentions a possible association with BMI for women's risk of heat stroke, and we think the paper you referred to is indirectly supportive of this (we added it to the cited papers). However, we do not have information other than the age and gender of the subjects, so it is not beyond speculation. We newly described this point (lines 226-228).
7. (Figure 1) What differentiates between dark areas (Yonago - Tottori) and light area (Kurayoshi) shown on the map?
-We simply color-coded the areas of every district. We added an explanation to the explanation in Figure 1.
8. Designating to Figure 2, it would be clear if the authors could describe the stratified analysis related to temperature classifications (<30°C; 30-34°C; 35-36°C; 37+°C).
Maybe this is an open question. I am a bit confused about how this study analyzes the risk of developing heat-stroke at temperatures ≥35°C, considering the results of the statistical analysis in Table 1 show that the highest temperature of the three studied regions is around 34°C (to be precise 33.5 C for the maximum temperature in Tottori).
-Thank you for your suggestion. From July to August, the percentage of days 35°C≤ is about 30% in Tottori and 25% in Yonago, which is not small. To avoid misunderstanding, we made a new table for the days with the minimum of 25°C≤ and maximum temperatures of 30°C≤/35°C≤ (Table 2).
9. (Lines 158-160) “Higher nighttime temperatures have been reported to lead to sleep deprivation, with the greatest impact seen in the summer months among low-income and elderly individuals”. Regarding this statement, is it possible to consider socioeconomic conditions (e.g. income, education, etc.) as the potential covariates for model adjustment related to the impact of air temperature on heat-stroke? Perhaps this can also be noted as a limitation of this study.
- As noted within "2.2. Ambulance Transport Data", we do not have data other than the age and gender of the cases. Considering the patient background you have pointed out, we think this would be a very comprehensive paper. However, this is not possible at this time, so we have added a note in the Limitations section (lines 250-251).
10. In strengthening the results of this study, the wet-bulb globe temperature (WBGT) index as a measure of heat stress can also be considered.
- Initially, we were also thinking of evaluating the effects of relative humidity and WBGT. As we described in “Discussion”, the reason why we did not use data other than temperature in this study is that in our previous study, there was no association between humidity and the number of heat stroke transports. Of the three weather observation sites, Kurayoshi in the central region is small in scale and does not record humidity data. Therefore, the main reason is that it is not possible to calculate indicators that includes humidity. We made additions and corrections to take these points into account (line 230-235).
--
Again, thank you for giving us the opportunity to strengthen our manuscript with your valuable comments and queries. We have worked hard to incorporate your feedback and hope that these revisions persuade you to accept our submission.
Sincerely yours,
Shinji Otani
Round 2
Reviewer 1 Report
The paper has been improved after the first round of reviews and most of my comments have been addressed in a satisfactory way. However, regarding the last point, I am still not totally convinced that the methodology used is the correct one. In fact the authors have replied that they did not consider relative humidity as one of the weather stations considered does not have instruments to measure the relative humidity. This is, of course, a constraint that cannot affect the whole study, as there are many alternative ways to measure or to calculate RH for the area considered (there are very cheap portable instruments that allow us to measure RH with enough precision). Moreover, the authors do not demonstrate (and this could be done by calibrating the data considering RH in the other two sites where RH measurements is available) that RH is not influential. More work is, therefore, still to be made before the paper will be ready to be published.
Reviewer 2 Report
Thank you for the improved version. I accept the publications if you correct this typos:
Line 22: Add "under the age of 18"
Line 38: Add blank space before parenthesis
Line 76: Change to "municipality"
Line 139+140: Change to "mAximum"